# Brachygnathia Inferior in Cloned Dogs Is Possibly Correlated with Variants of Wnt Signaling Pathway Initiators

**DOI:** 10.3390/ijms23010475

**Published:** 2022-01-01

**Authors:** Yong-ho Choe, Tai-Young Hur, Sung-Lim Lee, Seunghoon Lee, Dajeong Lim, Bong-Hwan Choi, Haeyun Jeong, Jin-Gu No, Sun A Ock

**Affiliations:** 1Department of Theriogenology and Biotechnology, College of Veterinary Medicine, Gyeongsang National University, Jinju 52828, Korea; yhchoego@gmail.com; 2Animal Biotechnology Division, National Institute of Animal Science (NIAS), Rural Development Administration (RDA), 1500, Kongjwipatjwi-ro, Isero-myeon, Wanju 55365, Korea; tyohur@korea.kr (T.-Y.H.); sage@korea.kr (S.L.); ahzk111@gmail.com (H.J.); shrkftm@gmail.com (J.-G.N.); 3Research Institute of Life Sciences, Gyeongsang National University, Jinju 52828, Korea; 4Division of Animal Genomics & Bioinformatics, NIAS, RDA, Kongjwipatjwi-ro, Isero-myeon, Wanju 55365, Korea; lim.dj@korea.kr (D.L.); bhchoi@korea.kr (B.-H.C.)

**Keywords:** cloned dog, brachygnathia inferior, whole-genome sequencing, Wnt signaling pathway

## Abstract

Abnormalities in animals cloned via somatic cell nuclear transfer (SCNT) have been reported. In this study, to produce bomb-sniffing dogs, we successfully cloned four healthy dogs through SCNT using the same donor genome from the skin of a male German shepherd old dog. Veterinary diagnosis (X-ray/3D-CT imaging) revealed that two cloned dogs showed normal phenotypes, whereas the others showed abnormal shortening of the mandible (brachygnathia inferior) at 1 month after birth, even though they were cloned under the same conditions except for the oocyte source. Therefore, we aimed to determine the genetic cause of brachygnathia inferior in these cloned dogs. To determine the genetic defects related to brachygnathia inferior, we performed karyotyping and whole-genome sequencing (WGS) for identifying small genetic alterations in the genome, such as single-nucleotide variations or frameshifts. There were no chromosomal numerical abnormalities in all cloned dogs. However, WGS analysis revealed variants of Wnt signaling pathway initiators (WNT5B, DVL2, DACT1, ARRB2, FZD 4/8) and cadherin (CDH11, CDH1like) in cloned dogs with brachygnathia inferior. In conclusion, this study proposes that brachygnathia inferior in cloned dogs may be associated with variants in initiators and/or regulators of the Wnt/cadherin signaling pathway.

## 1. Introduction

Animal cloning is a useful technology in developmental biology and genetic studies and in the restoration of endangered species [1,2,3]. Since the first successful dog cloning was reported [1], cloning has been applied not only in the commercial breeding of companion dogs but also in the production of working dogs with various desirable abilities [4].

Dog cloning differs from cloning in other animals, such as sheep, cattle, and pigs, owing to the different reproductive processes, such as ovulation of oocytes at the metaphase I stage. Therefore, dog cloning is performed with in vivo-matured oocytes, after which cloned embryos are quickly transferred into the oviduct to overcome the inadequate in vitro culture systems [5,6]. The cloned offspring are expected to not only genetically but also phenotypically identical to the original donor dog [7]. However, abnormal phenotypes not present in the original dog may appear in the clones, and this is presumed to occur during developmental events. This phenomenon has been commonly reported and studied in detail in other animals [2,8,9], but minimally reported in cloned dogs.

Defects in cloned animals mainly include high or low birth weight, placental abnormalities, and pulmonary and cardiovascular disorders [3]. Brachygnathia inferior (also called underbite, overshot, parrot mouth, or prognathia), an osteogenesis imperfecta presenting as shortening of the mandible, is a common congenital anomaly in sheep and cattle [10,11]. However, brachygnathia inferior is very rarely found in dogs [12], and there have been no reports of brachygnathia inferior in cloned dogs.

In the present study, we identified that out of four dogs cloned under the same conditions, except for the oocytes, two exhibited abnormalities including brachygnathia inferior. To the best of our knowledge, this is the first attempt to detect and determine the causes of brachygnathia inferior in cloned dogs. Therefore, we aimed to determine the genetic cause of brachygnathia inferior in these cloned dogs. Genome sequencing is a powerful tool for discovering genes and genetic variants that cause a disease [13]. Whole-genome sequencing (WGS) can provide information on the entire DNA sequence of the genome of an individual and serve as a tool for determining the genomic variation that increases the risk for common and rare disorders. Using WGS and functional prediction tools, we identified the specific genes that were upregulated only in these abnormal cloned dogs and the signaling pathways associated with the phenotypic features of brachygnathia inferior. We revealed mutations in the initiators and/or modulators of two important signaling pathways related to brachygnathia inferior in the cloned dogs.

## 2. Results

### 2.1. Production of Cloned Dogs

A total of 89 nuclear transfer (NT) embryos were transferred to 10 surrogate mother dogs. The pregnancy rate was confirmed to be 20% (2 out of 10 dogs). Surrogate mother dogs (SMD), called SMD1 and SMD2, gave birth to one offspring (NT-1) by cesarean section (≈60 days of gestation) and five offspring (NT-2 to -6) by natural delivery, respectively (Appendix A). Among the five offspring delivered by SMD2, three (NT-2, 3, and 4) survived, but two died: one was stillborn (NT-5), and the other died of hypothermia (NT-6) at one day after delivery (Figure 1). The cloning efficacy ratio, calculated from the number of live offspring per number of transferred embryos, was 5.6%.

To evaluate the genetic identity of the offspring, we compared canine-specific polymorphic microsatellites between the cloned puppies and donor cells. As shown in Table 1, the cloned puppies and donor cells showed genetic homogeneity, confirming that the puppies were cloned from the original dog.

### 2.2. Care and Feeding of Cloned Offspring

NT-1, the first cloned puppy, was fed by bottle because the surrogate mother did not care for her baby. During the artificial nursing period, slight pneumonia occurred, but was completely cured within one week. NT-2, -3, and -4, who were born by the same SMD2, were successfully breastfed by their SMD2. At the age of one month old, every puppy was stopped from milk feeding and fed commercial feed. The weight of all cloned puppies measured daily was in normal range, but the weight growth rates of NT-1 and -3 were lower than those of NT-2 and -4 until one month after birth (Appendix A). Both NT-1 and -3 showed jaw abnormalities, such as open-bite malocclusion of the mandible, starting at one month after birth, especially NT-1 (Figure 2A). Further detailed analyses were performed to determine the cause.

### 2.3. Clinical Diagnosis of Brachygnathia Inferior

First, NT-1, which presented severe jaw abnormality, was subjected to veterinary pathological analysis. Complete blood count (CBC) and biochemical parameters were within the reference ranges, without significant differences. CBC values were similar between the donor dog and the cloned offspring (Table 2). There were no differences in biochemical parameters, such as creatinine, glucose, blood urea nitrogen (BUN), gamma-glutamyl transferase (GGT), albumin (ALB), total bilirubin (TB), total protein (TP), alanine aminotransferase (ALT), aspartate aminotransferase (AST), creatine kinase, cholesterol, and amylase levels, between the original dog and the clones (Table 3).

The cloned dogs were diagnosed by visual inspection and X-ray/computed tomography (CT) examinations. Mandibular malocclusion was observed in both NT-1 and -3, whereas NT-1 presented severe mandibular malocclusion. In the case of NT-1, malalignment of the central axis of the skull was confirmed in the dorsoventral view of the craniofacial radiograph (Figure 2B(i)). Additionally, the craniofacial angle between the maxilla and the mandible in the lateral view of the craniofacial region was measured by radiography (Figure 2B(ii)). NT-1 showed an increased craniofacial angle compared to the donor dog and NT-2. NT-1 showed normal teeth arrangement in terms of the number and order, but its tooth morphology was irregular and denser compared with that of the donor dog and NT-2. In NT-1, it was confirmed through three-dimensional CT images that the maxillary canines protruded to a greater extent than the mandibular canines (Figure 2C), unlike those in the donor dogs and NT-2. When the above findings were considered, NT-1 was diagnosed with typical brachygnathia inferior.

### 2.4. Chromosomal Aberrations in the Donor and Cloned Dogs

Karyotyping was performed to analyze chromosomal abnormalities in peripheral blood samples from all cloned dogs and the donor dog. All samples were read as normal diploids with 78 + XY (Figure 3). It was found that cloning did not induce chromosomal aberrations, observed as numerical and structural abnormalities, and thus brachygnathia inferior did not occur as a result of large-scale chromosomal aberrations.

Therefore, further studies were performed to identify the cause of brachygnathia inferior in the cloned dogs on the basis of single nucleotide variations (SNVs) or short insertions/deletions (indels).

### 2.5. Identification and Validation of Candidate Genes for Brachygnathia Inferior by Whole-Genome Sequencing

WGS was performed for the donor dogs and cloned offspring, and high-quality sequence data were obtained. Data on variants were subjected to quality control, and the results are presented in Appendix A. The circos plot of WGS confirmed the presence of genomic variations, including SNVs and indels, between the groups (Figure 4). Coexisting variants present in all animal subjects including the donor dog were filtered. Through Venn diagram analysis, we identified 10,112 variants in 3164 genes, including unique SNVs and indels, exclusively in the group with brachygnathia inferior (Figure 4B). These variants were located in the protein-coding and intergenic regions of the 3164 genes.

Next, we examined whether these 3164 genes have phenotype-related functions. To determine the biological characteristics of these candidate genes for brachygnathia inferior, we performed Gene Ontology (GO) analysis on biological processes using the DAVID database. Among the 3164 genes, 1471 genes were significantly enriched (*p* < 0.001), as shown in Figure 5A. The biological functions of these genes are mostly related to cellular and systemic developmental processes. Interestingly, out of the 1471 genes, 221 were involved in anatomical structure morphogenesis (*p* = 0.000007).

Furthermore, functional prediction was conducted for the 3164 candidate genes using the Protein Analysis Through Evolutionary Relationships (PANTHER) annotation system. Out of the 3164 candidate genes, 1913 were mapped on 110 pathways, and the top 10 pathways are presented in Figure 5B. The top four enriched pathways were identified as the Wnt (47%), cadherin (31%), integrin signaling (30%), and gonadotropin-releasing hormone receptor (27%) pathways.

A Venn diagram showing the overlapping of genes related to the four pathways revealed that the Wnt (Figure 5C(ii)) and cadherin (Figure 5C(iii)) signaling pathways had many shared genes, compared to the integrin (Figure 5C(i)) and gonadotropin (Figure 5C(iv)) signaling pathways. Thus, 50 candidate genes for brachygnathia inferior were extracted from the 1913 genes related to the Wnt/cadherin signaling pathway (Figure 5D, detailed in Table 4). Of these 50 genes, 30 were shared between the Wnt and cadherin signaling pathway.

Detailed information on mutations in 50 candidate genes for brachygnathia inferior is summarized as shown in Table 4. Especially, two uncharacterized proteins and six genes (CDH8, CDH12, PCDH9, CTNND2, PCDH9, and ENSCAFG00000023180) with more than 10 variations were identified. Thus, it is presumed that these specific variants in NT-1 and -3 cause alterations in genes related to the Wnt/cadherin signaling pathway, although the exact mechanism is unknown.

### 2.6. Interactive Network Analysis of Candidate Genes for Brachygnathia Inferior

To elucidate how these candidate genes for brachygnathia inferior interact with each other, we predicted a protein–protein interaction network using the STRING database. We found 132 interaction edges with an enrichment *p*-value < 10^−15^, as shown in Figure 6. A total of 50 candidate genes for brachygnathia inferior were confirmed to be related to the Wnt (yellow highlight)/cadherin (red highlight) signaling pathway. The network showed that the genes were closely interacting with each other, with WNT5B, ARRB2, CTNNA3, and CTNND2 at the center.

## 3. Discussion

Many animals have been cloned via nucleus transfer from somatic cells to mature oocytes in vitro [14,15,16,17,18]. However, owing to inadequate in vitro culture conditions and different estrus cycles, dogs are typically cloned using surgically recovered mature oocytes in vivo [5,6]. Nevertheless, the cloned dogs were also found to have many abnormalities as in other cloned animals [19]. In cloned animals, the causes of these abnormalities include incomplete reprogramming and imprinting, as well as inappropriate culture environment [20,21,22]. However, the cause has not been clearly identified. We found brachygnathia inferior for the first time in cloned dogs and attempted to identify the cause through analysis of genetic signals involved in embryonic fate.

The efficiency of cloning, which is calculated from the number of viable offspring per transferred embryo, is known to be 5–15%, depending on the animal species [3]. The cloning efficiency of the first cloned sheep, Dolly, was 3.4%. In the present study, the 5.6% cloning efficiency was similar to that reported in previous studies. This efficiency value indicates that 95–85% of SCNT embryos died before reaching full term [23]. In addition, a high incidence of malformations, large offspring syndrome, placental defects, and brain defects, as well as pulmonary, renal, and cardiovascular failure were observed in the placenta, fetuses, and offspring in cloned animals [3,22]. Incomplete remodeling and abnormal epigenetic modification of somatic nucleic acids have been identified as the cause of these abnormalities [20,21,22]. In cloned dogs, cleft palate and abnormal external genitalia such as failure of preputial closure at the ventral distal part and persistent penile frenulum have been reported [19]. In the present study, although there were no abnormalities in hematological parameters and chromosome numbers among the four cloned dogs produced by SCNT under the same environmental conditions, brachygnathia inferior caused by growth failure of mandible occurred in two dogs, as determined by morphological observation, X-ray imaging, and CT diagnosis at 1 month after birth.

Craniofacial malformations, such as cleft palate and mandibular abnormalities, have been studied in many animals. Despite the relatively high incidence of these disorders, their genetic cause have not been studied in detail. In cattle, trisomy 17 and 22 [24,25], as well as mutations in GON4L [26], are related to brachygnathia inferior. In sheep, frameshift in OBSL1 has been shown to affect brachygnathia inferior [27], and mutations in COL1A1 and COL1A2 cause osteogenesis imperfecta [28]. In dogs, LINE-1 insertion within DLX6 induces cleft palate and mandibular abnormalities, as reported by a genome-wide association study in a canine model [29]. The skull shape of dogs is regulated by a missense mutation in BMP3 [30]. In the present study, WGS analysis revealed that cloned dogs with brachygnathia inferior had 10,112 SNVs and indels in 3164 genes, compared to normal dogs without brachygnathia inferior. Interestingly, two variants between BMP3 and PRKG3, n.5244256C>T and n.5248552A>G, were detected in cloned dogs with brachygnathia inferior. However, these two single-nucleotide variants were located in the intergenic region. It was difficult to evaluate their effect on mandibular abnormalities [31].

The Wnt signaling pathway is known to be involved in cell destiny, polarity, and migration during embryonic development and differentiation [32,33]. The major Wnt signaling pathways are divided into canonical pathways that start with the binding of Wnt ligands (WNT 1, WNT 3, WNT 7, etc.), Frizzled (Fzd) receptors, and low-density lipoprotein receptor-related protein (LRP) 5 or LRP6, as well as into non-canonical pathways that start with the binding of WNT5a class ligand and FZD receptor. The non-canonical Wnt pathway consists of two types: the planar cell polarity (PCP) pathway, related to cell polarity and migration, and the Wnt/Ca^2+^ pathway, which is involved in the activation of Ca^2+^-dependent proteins (CaMK2, PKC, and calcium) related to cell differentiation, relocation, and adhesion. In both canonical and non-canonical Wnt signaling pathways, the cascade begins through the activation/induction of Dishevelled (Dsh) via the binding of Wnt and FZD [34,35,36]. In the present study, through Wnt/cadherin signaling network analysis in cloned dogs with brachygnathia inferior, we found that WNT5B interacted closely with FZD4, FZD8, DVL2, β-adrrestin2 (ARBB2), and DVL binding antagonist of β-catenin (DACT1) as negative regulators of WNT signaling, and that these genes were closely related to initiators of canonical and non-canonical Wnt signaling pathways [37]. Pathogenic variants of these genes were also identified, including tumor or posterior malformation in FZD4 (c.74G>T), ARRB2 (c.287A>C), and DACT1 (c.1561-1G>A) in humans and mice [37,38,39].

In the β-catenin-independent pathway, the WNT5 subfamily, including WNT5a and WNT5b, which can combine with FZD4 or FZD8, is closely involved in osteogenesis, and disruption of the WNT5 subfamily leads to skeletal defects [40,41]. In osteoblastic cell lines, cadherin 11 (calcium-dependent adhesion, CDH 11) is implicated in bone development and maintenance [42]. E-cadherin (CDH1) interacts with catenin α-δ types in the cytoplasm, and their complexes are important for epithelial cell polarity and function [43]. Therefore, we propose that brachygnathia inferior in cloned dogs was affected by two pathways: (1) the non-canonical WNT pathway, such as activation by DVL2/ARRB2 or inhibition by DVL2/DACT1 after the binding of WNT5b to FZD4/FZD8, and (2) the catenin/cadherin pathway via the interaction of α/δ catenin (CTNNA3, CTNND2) and CDH by ARRB2. The involvement of the catenin/cadherin pathway is supported by the discovery of LOC489647 (cadherin-1-like) [44].

Although it is not possible to precisely estimate how the Wnt and cadherin signaling pathways are differentially expressed in dogs cloned under the same conditions, it is presumed that the use of oocytes recovered from different dogs affects the reprogramming of donor somatic cell nuclei from the original donor dog. This hypothesis is supported by previous reports that oocyte cytoplasm extracts such as ooplasmic factor can regulate the epigenetic reprogramming of somatic cell nuclei, such as the demethylation of histones [45,46].

## 4. Conclusions

This study revealed that brachygnathia inferior in cloned dogs was associated with variants in the initiators and/or regulators of the Wnt/cadherin signaling pathway, especially the non-canonical Wnt signaling pathway via WNT5b. Although the direct cause of the abnormalities in cloned dogs, such as brachygnathia inferior, could not be determined, it was presumed that the oocytes used for cloning altered the reprogramming of the donor somatic cells. In order for this hypothesis to be proven, further gene editing and epigenetic reprogramming error studies are necessary in order to identify abnormalities in cloned offspring. However, considering that dogs are companion animals, and not laboratory animals such as mice, future research for identifying the related genetic variants should utilize genetic samples from dogs with brachygnathia inferior that are naturally born.

## 5. Materials and Methods

### 5.1. Animals

All experiments were authorized by the Animal Center for Biomedical Experimentation at the National Institute of Animal Science of the Rural Development Administration (approval number 2015-143 on 21 May 2015) and followed animal care and use guidelines.

### 5.2. Cloning of Dogs

For dog cloning, somatic cell nuclear transfer (SCNT) and embryo transfer were performed according to a previously described protocol [6] with minor modifications, as shown in Figure 1. As donor cells, ear fibroblasts were collected from a 5-year-old male German shepherd dog (original donor) via ear skin biopsy after anesthesia. The original dog was diagnosed with a normal phenotype that was clinically healthy and without physical disabilities. Fibroblasts at the second to third passage were stored in LN_2_ before use as donor cells in SCNT. Three days prior to SCNT, the cryopreserved cells were thawed and cultured at a seeding concentration of 5 × 10^4^ cells per well in a 4-well dish. Cells at the second to third passage were used for the production of cloned embryos. After SCNT, the embryos were immediately surgically transferred into the oviduct of a surrogate mother using a previously described method [6]. Pregnancy rates were determined by ultrasound diagnosis at ≈31 days after embryo transfer.

### 5.3. Microsatellite Analysis for Confirmation of Paternity

For confirmation of genetic identity, genomic DNA was extracted from the blood of four cloned dogs with a Wizard Genomic DNA Purification Kit (Promega, Madison, WI, USA) and from donor cells with PureLink™ Genomic DNA Mini Kit (Thermo Fisher Scientific, Carlsbad, CA, USA). Multiplex PCR was performed using the GeneAmp PCR system 9700 (ABI) using a previously reported method (Ko et al., 2019). PCR products were analyzed using a DNA sequencer (ABI 3730xl; Applied Biosystems, Foster City, CA, USA), and microsatellite analysis was conducted using GeneSMapper version 4 (ABI). Microsatellite markers, such as FH2537, FH3005, FH3372, FH3116, REN51C16, REN2770O5, FH2834, REN204K13, FH2097, FH2712, and FH2998, were selected according to previous studies [47,48].

### 5.4. Hematological and Biochemical Analysis of Blood

Blood samples were collected from each dog via jugular venipuncture. For complete blood count (CBC) measurement, blood samples were collected into EDTA-containing tubes, and leukocytes, erythrocytes, and thrombocyte were counted using an automated hematology cell counter (MS9-5V; Melet Schloesing Lab, Osny France). To assess kidney, liver, and heart functions, we performed blood chemistry analysis using a bench-top dry chemistry analyzer (Vettest 8008 Chemistry Analyzer; IDEXX Lab, Chalfont St Peter, United Kingdom), in which creatinine, glucose, blood urea nitrogen (BUN), gamma-glutamyl transferase (GGT), albumin, total bilirubin, total protein (TP), alanine aminotransferase (ALT), aspartate aminotransferase (AST), creatine kinase, cholesterol, and amylase levels were determined.

### 5.5. X-ray and CT Imaging

X-ray and CT imaging were performed using a two-channel multi-detector row CT scanner (Somatom Emotion, Siemens Medical System, Erlangen, Germany). For CT scanning, the animals were anesthetized by inhalation of 2% isoflurane, and CT was performed at 110 kV, 36 mAs, and 1 mm slice thickness. Datasets were transferred to a separate workstation, and the volume and size of the vertebral window by pediculectomy for each site were measured using the Lucion software (Infinitt Technology, Seoul, Korea).

### 5.6. Karyotype Analysis

For chromosome analysis, peripheral blood samples were added to RPMI media (1640; Gibco, Rockville, MD, USA) supplemented with FBS and phytohemagglutinin, and cultured overnight into CO_2_ incubator at 37 °C. The blood cells were arrested in metaphase by adding 0.1 µg/mL of colcemid for 1 h, and then harvested using 0.25% trypsin/EDTA solution. The single-cell suspension was incubated in hypotonic solution buffer (0.075 M KCl) for 45 min and fixed with methanol-acetic acid (3:1). After fixation, condensed chromosomes were spread on pre-cleaned glass slides and stained with Giemsa solution. Karyotyping of cultured cells was performed using standard cytogenetic techniques, revealing a female chromosomal constitution of 2n = 78, XY.

### 5.7. Whole-Genome Sequencing, Sequence Mapping, and Variant Calling

Blood samples were collected from the normal phenotype group (original dog, NT-2, and NT-4) and the brachygnathia inferior group (NT-1 and NT-3), and genomic DNA was extracted using the TruSeq Nano DNA Sample Prep Kit (Illumina, San Diego, CA, USA). Whole-genome sequencing was performed using the Illumina HiSeq 2500 sequencing platform (Illumina, San Diego, CA, USA). Skewer software (v0.2.2) was used for adapter trimmer, and BWA (v0.7.15) were used for aligning the collected sequence data to the canine reference genome (CanFam 3.1). The Genome Analysis Toolkit (GATK, v2.3.9Lite) was used for improvement of alignment errors and genotype calling and refining with default parameters. SNP-calling procedure was performed to discover SNPs using SAM tools (v.1.3.1). The detected SNPs were then annotated to functional categories using SnpEff software (v4.3a).

### 5.8. Predictive Functional and Interaction Analyses of Brachygnathia Inferior Candidate Genes

To validate the basic biological function of brachygnathia inferior candidate genes, we mapped a list of genes to the biological process (BP) of Gene Ontology (GO) in the DAVID database (v.6.8, accessed on 1 January 2019, http://david.abcc.ncifcrf.gov). Significantly enriched GOBP categories of brachygnathia inferior candidate genes were determined by the enrichment *p*-value. Signaling pathway enrichment analysis was performed using PANTHER pathway analysis tools (v.14.0, accessed on 1 January 2019, http://pantherdb.org). The signaling pathway of brachygnathia inferior candidate genes was determined by the number of genes mapped on each pathway and the percentage of enriched gene number against the total number of each pathway component genes.

To analyze the gene-to-gene functional correlation of brachygnathia inferior candidate genes, we constructed an interaction network using Search Tool for the Retrieval of Interacting Proteins (STRING, accessed on 1 January 2019, http://string-db.org).

## Figures and Tables

**Figure 1 ijms-23-00475-f001:**
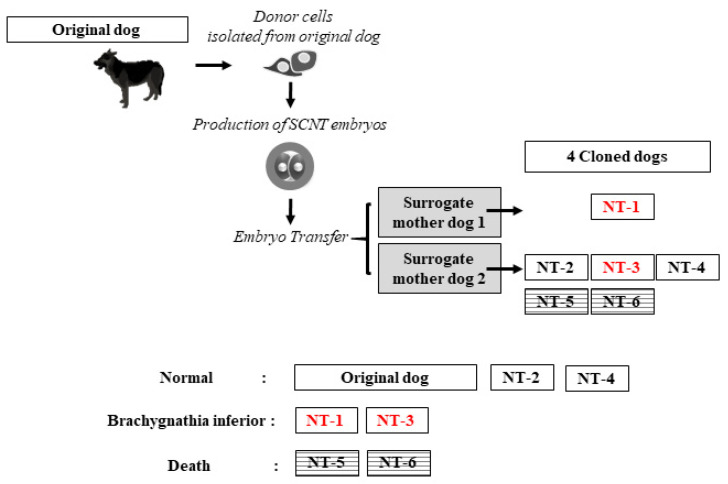
Schematic description of the dog cloning process via somatic cell nuclear transfer (SCNT), showing the descendant pedigree and phenotype of offspring. Donor cells were collected from a 5-year-old male German shepherd dog (original donor). NT refers to the cloned offspring produced via SCNT. Red colors (NT-1 and NT-2) were represented cloned dogs with brachygnathia inferior.

**Figure 2 ijms-23-00475-f002:**
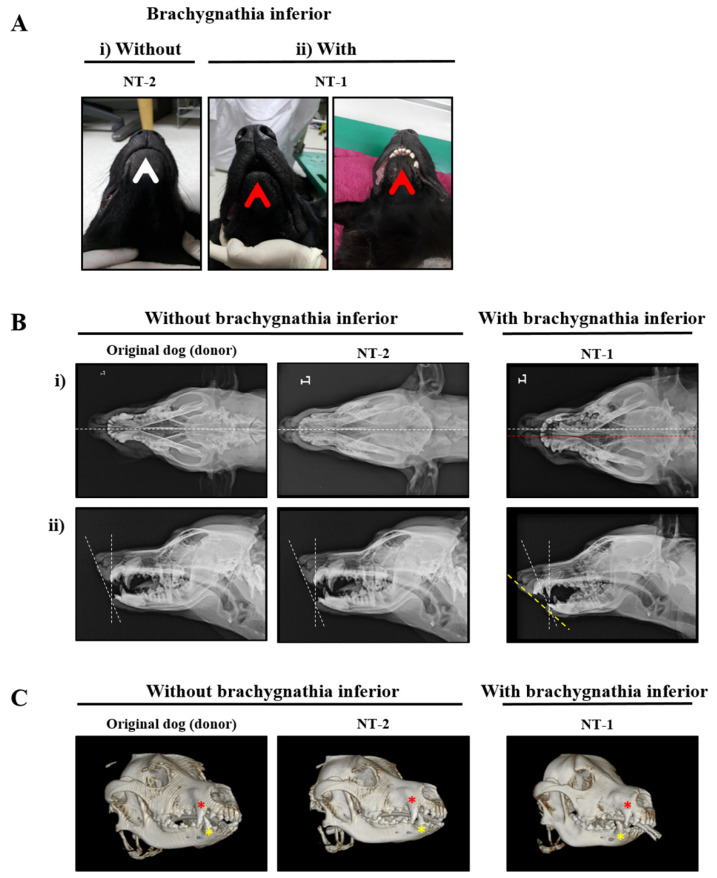
Clinical diagnosis of cloned dogs. (**A**) Visual inspection of mandibular malocclusion in the cloned dogs. (**i**) A cloned dog without brachygnathia inferior (NT-2); (**ii**) a cloned dog with brachygnathia inferior (NT-1). Arrows indicate the lower jaw with (red) and without (white) brachygnathia inferior. (**B**) Craniofacial radiographic images of the cloned dogs. Left column, the original dog as a control; central column, a cloned dog without brachygnathia inferior (NT-2); right column, a cloned dog with brachygnathia inferior (NT-1). (**i**,**ii**) are shown in the dorsoventral and lateral view of their craniofacial profile, respectively. White dotted lines in (**i**) indicate the central axis of the skull in the dorsoventral view of the craniofacial profile. The cross of white and yellow dotted lines in (**ii**) indicate the craniofacial angle between the maxilla and mandible. (**C**) Three-dimensional volume-rendered computed tomography images. Left, the original dog as a control; center, a cloned dog without brachygnathia inferior (NT-2); right, a cloned dog with brachygnathia inferior (NT-1). Red and yellow asterisks indicate the maxillary and mandible canines, respectively.

**Figure 3 ijms-23-00475-f003:**
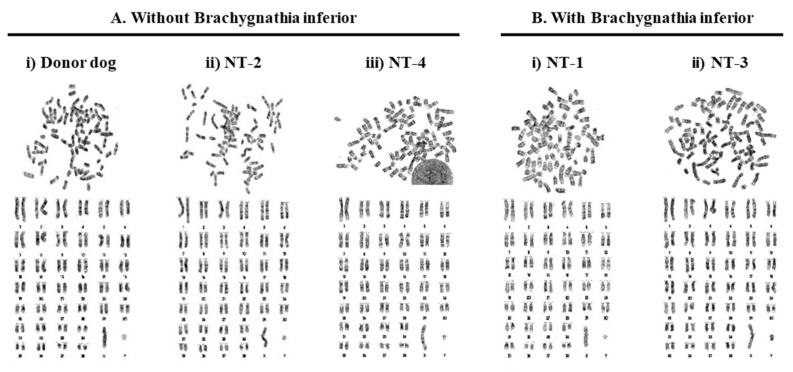
Karyotyping in cloned dogs. Karyotyping was performed using peripheral blood samples from the dogs. The normal diploid chromosome number for dogs is 78, with the autosomes acrocentric, whereas the X and Y chromosomes are the large and small submetacentric chromosomes, respectively. (**A**,**B**) Dogs with and without brachygnathia inferior, respectively. Aa and Ab represent the original dog (as control, donor) and the cloned dog, respectively. All dogs were male and had a normal number of chromosomes.

**Figure 4 ijms-23-00475-f004:**
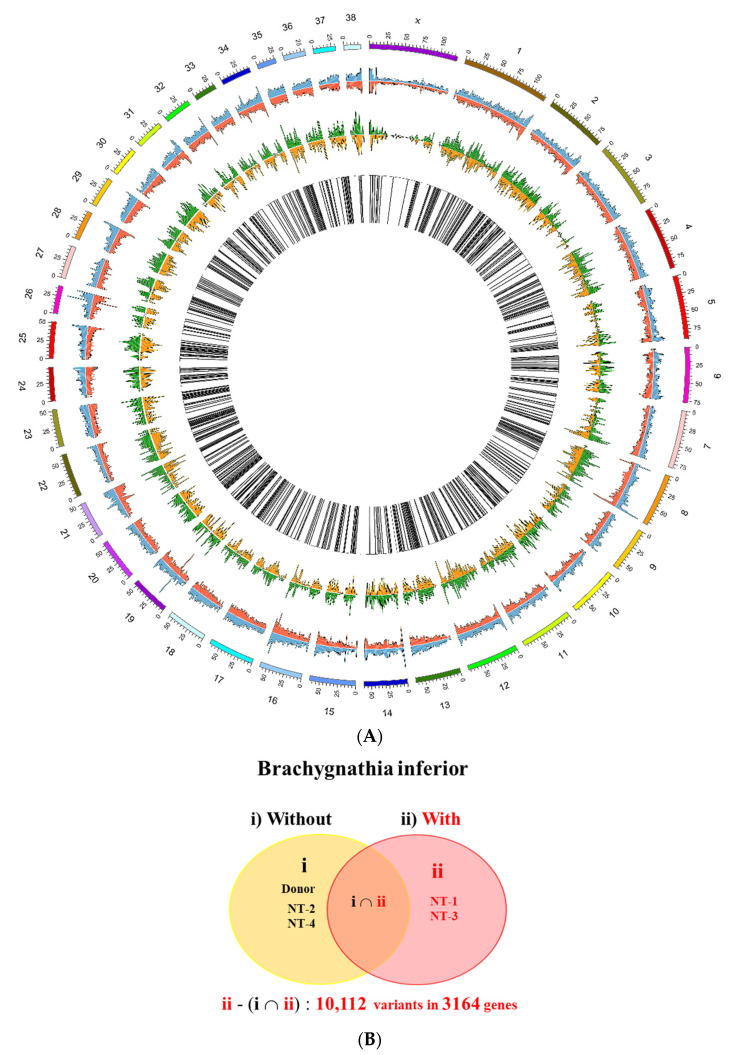
Comparative analysis of whole-genome sequences between the original dog (donor) and cloned dogs. (**A**) Circos plot comparing variants in genome sequence between the cloned dogs with (NT-1 and -3) and without brachygnathia inferior (original dog, NT-2 and -4). From the outside, each layer indicates reference chromosomes, the number of single-nucleotide variants (SNVs) (blue: normal, red: affected), and the number of insertions and deletions (indels) (green, normal; orange, affected). The black bar represents the differences between the normal and affected samples. (**B**) Venn diagram of specific variants between the dogs with (**ii**) and without (**i**) brachygnathia inferior.

**Figure 5 ijms-23-00475-f005:**
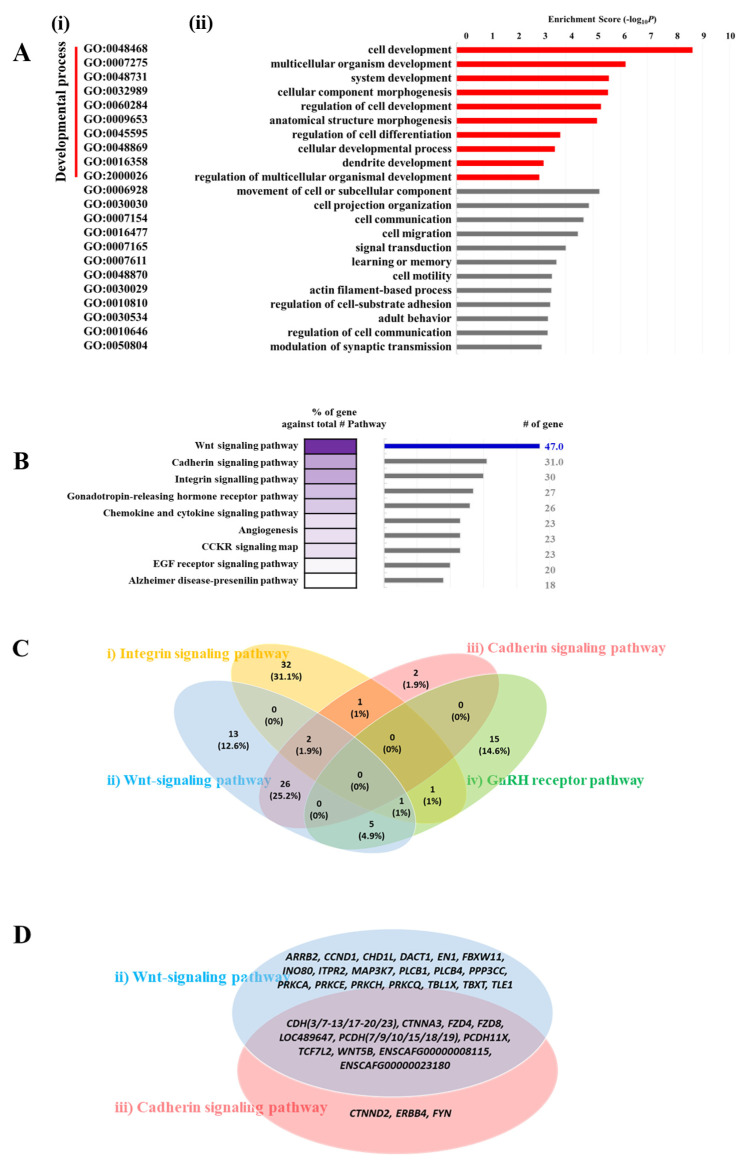
Gene ontology (GO) term enrichment analysis and Protein Analysis Through Evolutionary Relationships (PANTHER, v.14.0, 1 January 2019, http://pantherdb.org). The 3164 genes with specific variants found in cloned dogs with brachygnathia inferior were included in these analyses. (**A**) Classification of 1471 genes according to the GO biological process using Database for Annotation, Visualization, and Integrated Discovery (DAVID, v.6.8, 1 January 2019, http://david.abcc.ncifcrf.gov). (**i**,**ii**) The categories and bar plots of the GO biological processes, respectively. The red lines in “a” represent GO categories that participate in the development process. The *p*-values of each process were converted to −log10 P to calculate the enrichment score. (**B**) Mapping of 1913 genes via PANTHER. Relative gradient violet color represents the percentage of the enriched gene number relative to the total number of each pathway component gene. The bar plot displays the number of enriched genes. Venn diagrams in (**C**,**D**) represent the number of overlapping genes of the top-four most enriched pathways (**i**–**iv**) and the common and different gene list between the top-two gene pathways (**ii**,**iii**), respectively. (**i**–**iv**) Integrin (**i**), Wnt (**ii**) and cadherin (**iii**) signaling pathways, and the GnRH receptor pathway (**iv**).

**Figure 6 ijms-23-00475-f006:**
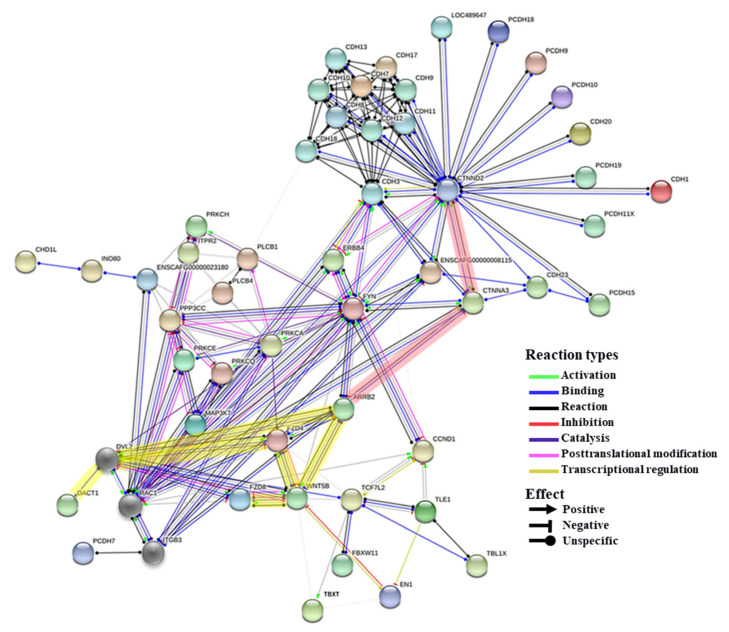
Interactive network of candidate genes for brachygnathia inferior. Each edge indicates interaction between two genes. The interaction types and their effects are described in the figure. This analysis was performed using the Search Tool for the Retrieval of Interacting Genes/Proteins (STRING) database. Yellow and red letters indicate genes involved in the Wnt and cadherin pathways, respectively.

**Table 1 ijms-23-00475-t001:** Matching of microsatellite of between donor cells and cloned offspring.

Source	FH2537		FH3005		FH3372		FH3116		REN51C16		REN2770O5		FH2834		REN204K13		FH2097		FH2712		FH2998	
Donor cells	146	146	224	224	154	158	190	190	255	259	333	333	265	265	248	248	284	288	174	174	208	228
NT-1	146	146	224	224	154	158	190	190	255	259	333	333	265	265	248	248	284	288	174	174	208	228
NT-2	146	146	224	224	154	158	190	190	255	259	333	333	265	265	248	248	284	288	174	174	208	228
NT-3	146	146	224	224	154	158	190	190	255	259	333	333	265	265	248	248	284	288	174	174	208	228
NT-4	146	146	224	224	154	158	190	190	255	259	333	333	265	265	248	248	284	288	174	174	208	228

Donor cells used for nuclear transfer (NT); cloned offsprings (NT-1 to NT-4) were produced by NT.

**Table 2 ijms-23-00475-t002:** The values of complete blood counts for original donor and cloned dogs.

Parameters (Unit)	Original Dog (Donor Cells)	Brachygnathia Inferior		
		without		with
		NT2	NT4	NT-1
RBC (10^12^/L)	8.85	5.9	6.74	7.43
Hematocrit [Hct] (%)	56.7	37.7	41.2	48.1
Hemoglobin [Hb] (g/dL)	19.6	12.1	14.3	15.8
MCV (fL)	64.1	63.9	61.1	64.7
MCH (pg)	22.1	20.5	21.2	21.3
MCHC (g/dL)	34.6	32.1	34.7	32.8
PDW (%)	20.2	18.7	19.5	18.9
Reticulocyte (%)	0.2	1.8	0.5	1.1
Reticulocyte (10^3^/uL)	18.6	108	30.3	81.7
WBC (10^9^/L)	13.8	15.7	19.2	13.86
WBC-Neut (%)	70.6	56.9	64.6	60
WBC-Lymph (%)	15.3	29.5	24.9	25.2
WBC-Mono (%)	7.8	8.3	8.7	7
WBC-Eos (%)	6.2	5.2	1.7	7.8
WBC-Baso (%)	0.1	0.1	0.1	0
WBC-Neut (10^9^/L)	9.75	8.94	12.4	8.32
WBC-Lymph (10^9^/L)	2.11	4.63	4.76	3.49
WBC-Mono (10^9^/L)	1.08	1.31	1.66	0.97
WBC-Eos (10^9^/L)	0.85	0.81	0.33	1.08
WBC-Baso (10^9^/L)	0.01	0.01	0.02	0
Platelet (10^9^/L)	201	411	325	429
MPV (fL)		12.3	13.1	13.1
RDW (fL)		19.36	18.8	18.8
PCT (%)		0.5	0.56	0.56

RBC, red blood cell; MCV, mean corpuscular volume; MCH, mean corpuscular hemoglobin; MCHC, mean corpuscular hemoglobin concentration; PDW, platelet distribution width; WBC, white blood cell; MPV, mean platelet volume; RDW, red cell distribution width; PCT, plateletcrit.

**Table 3 ijms-23-00475-t003:** The values of blood chemistry parameters of original donor and cloned dogs.

Parameters (Unit)	Original Dog (Donor Cells)	Brachygnathia Inferior			
		without		with	
		NT2	NT4	NT-1	NT-3
Glucose (mg/dL)	68	31	104	62	90
BUN (mg/dL)	13	13	19	13	4
Creatinine (mg/dL)	1.2	1	0.7	1.1	0.3
BUN: Creatinine (Ratio)	11	13	25	11	12
Phosphorus-Inorganic (mg/dL)	3	8	8.7	5.7	8.3
Calcium (mg/dL)	10.8	11	9.9	10.9	11
Protein-Total (g/dL)	7.8	5.7	5	7.1	6.1
Albumin (g/dL)	4	2.9	2.7	3.4	2.9
Globulin (g/dL)	3.8	2.8	2.3	3.7	3.2
A/G ratio	1.1	1.1	1.2	0.9	0.9
ALT (U/L)	47	22	18	23	67
ALKP (U/L)	17	173	151	106	216
GGT (U/L)	0	0	2	0	8
Bilirubin-Total (mg/dL)	0.2	0.1	0.1	0.1	0.7
Cholesterol-Total (mg/dL)	191	119	171	170	237
Amylase (U/L)	727	539	299	863	323
Lipase (U/L)	341	331	352	322	93
Na+ (mmol/L)	153	150	147	151	140.9
K+ (mmol/L)	4.2	5	5.7	4.3	5.05
Na+:K+ (Ratio)	36	30	26	36	23.5
Cl^−^ (mmol/L)	110	107	107	107	111.3
Osmorality	301	294	296	297	

BUN, blood urea nitrogen; A/G ratio, the ratio of albumin and globulin; ALT, alanine aminotransferase; ALKP, alkaline phosphatase; AST, aspartate aminotransferase; GGT, gamma-glutamyltransferase.

**Table 4 ijms-23-00475-t004:** Variants found in WNT/cadherin signaling pathway in cloned dogs with brachygnathia inferior.

#	Gene Symbol (Gene Name)	Variant Information									
		CHR	POS	REF	ALTS	Putative Impact	Genotype				
							Without Brachygnathia Inferior			With Brachygnathia Inferior	
							Original Dog	NT-2	NT-4	NT-1	NT-3
1	ARRB2 (arrestin beta 2)	5	31834580	T	G	^+^ MOD	0/0	0/0	0/0	0/1	0/1
2	CCND1 (G1/S-specific cyclin-D1)	18	48505585	G	T	MOD	0/0	0/0	0/0	0/1	0/1
			48511836	T	TG	MOD	0/1	0/1	0/1	1/1	1/1
3	CDH3 (cadherin 3)	5	80888237	C	G	MOD	0/0	0/0	0/0	0/1	0/1
			80917292	A	T	MOD	0/0	0/0	0/0	0/1	0/1
4	CDH7 (cadherin 7)	1	11751189	C	T	MOD	0/0	0/0	0/0	0/1	0/1
			11765372	CATATATATATATATATAT	C	MOD	0/1	0/1	0/1	1/1	1/1
5	CDH8 (cadherin 8)	5	86401758	A	T	MOD	0/0	0/0	0/0	0/1	0/1
			86708399	C	A	MOD	0/0	0/0	0/0	0/1	0/1
			86742629	C	T	MOD	0/0	0/0	0/0	0/1	0/1
			86745597	T	TATACAC	MOD	0/1	0/1	0/1	1/1	1/1
			87225311	T	A	MOD	0/0	0/0	0/0	0/1	0/1
			87225313	T	A	MOD	0/0	0/0	0/0	0/1	0/1
			85863919	A	T	MOD	0/0	0/0	0/0	0/1	0/1
			86114270	C	CT	MOD	0/1	0/1	0/1	1/1	1/1
			86222689	T	G	MOD	0/0	0/1	0/0	1/1	1/1
			86296806	C	G	MOD	0/1	0/1	0/1	1/1	1/1
			86390011	TACCCC	T	MOD	0/1	0/1	0/1	1/1	1/1
6	CDH9 (cadherin 9)	4	78675775	A	T	MOD	0/0	0/0	0/0	0/1	0/1
			78830300	A	T	MOD	0/0	0/0	0/0	0/1	0/1
			78840862	T	C	MOD	0/0	0/0	0/0	0/1	0/1
			79012127	G	A	MOD	0/0	0/0	0/0	0/1	0/1
7	CDH10 (cadherin 10)	4	81023137	T	G	MOD	0/0	0/0	0/0	0/1	0/1
			81051478	A	T	MOD	0/0	0/0	0/0	0/1	0/1
			80613492	GTC	G	MOD	0/0	0/0	0/0	0/1	0/1
			80892138	GT	G	MOD	0/1	0/1	0/1	1/1	1/1
			80904036	A	T	MOD	0/0	0/0	0/0	0/1	0/1
8	CDH11 (cadherin 11)	5	84404903	AAGAG	AAGAGAG,A	MOD	0/0	0/0	0/0	0/2	0/2
			84473862	C	T	MOD	0/0	0/0	0/0	0/1	0/1
9	CDH12 (cadherin 12)	4	82754279	T	TATATATAG	MOD	0/1	0/1	0/1	1/1	1/1
			82856048	C	A	MOD	0/0	0/0	0/0	0/1	0/1
			82947779	G	A	MOD	0/0	0/0	0/0	0/1	0/1
			83011986	C	CGT	MOD	0/1	0/1	0/1	1/1	1/1
			83066202	G	T	MOD	0/0	0/0	0/0	0/1	0/1
			83101151	C	T	MOD	0/0	0/0	0/0	0/1	0/1
			81762398	T	A	MOD	0/0	0/0	0/0	0/1	0/1
			81811393	TTGAAA	T	MOD	0/1	0/1	0/1	1/1	1/1
			81895595	G	C	MOD	0/1	0/1	0/1	1/1	1/1
			81895601	C	A	MOD	0/1	0/1	0/1	1/1	1/1
			81934549	T	A	MOD	0/0	0/0	0/0	0/1	0/1
			81942964	A	C	MOD	0/0	0/0	0/0	0/1	0/1
			82120066	A	G	MOD	0/0	0/0	0/0	0/1	0/1
			82213787	AT	A	MOD	0/0	0/0	0/0	0/1	0/1
			82245479	C	CT	MOD	0/0	0/0	0/0	0/1	0/1
			82346773	T	G	MOD	0/1	0/1	0/1	1/1	1/1
			82391279	GT	G	MOD	0/0	0/0	0/0	0/1	0/1
			82427446	T	G	MOD	0/0	0/0	0/0	0/1	0/1
			82427448	T	G	MOD	0/0	0/0	0/0	0/1	0/1
			82567248	G	A	MOD	0/0	0/0	0/0	0/1	0/1
			82591151	C	T	MOD	0/0	0/0	0/0	0/1	0/1
			82672898	T	G	MOD	0/0	0/0	0/0	0/1	0/1
10	CDH13 (cadherin 13)	5	68789330	G	A	MOD	0/0	0/0	0/0	0/1	0/1
			69134746	GT	G	MOD	0/1	0/1	0/1	1/1	1/1
11	CDH17 (cadherin 17)	29	38980475	A	C	MOD	0/0	0/0	0/0	0/1	0/1
			38980931	C	T	MOD	0/0	0/0	0/0	0/1	0/1
12	CDH18 (cadherin 18)	4	84572993	G	T	MOD	0/0	0/0	0/0	0/1	0/1
			84609771	A	C	MOD	0/0	0/0	0/0	0/1	0/1
			84777718	CT	C	MOD	0/1	0/1	0/1	1/1	1/1
			84815886	CT	C	MOD	0/0	0/0	0/0	0/1	0/1
			84845137	AT	A	MOD	0/1	0/1	0/1	1/1	1/1
			84917669	TTA	T,TTATA	MOD	0/2	0/2	0/2	2/2	2/2
			85340830	G	A	MOD	0/0	0/0	0/0	0/1	0/1
13	CDH19 (cadherin 19)	1	11213176	A	T	MOD	0/0	0/0	0/0	0/1	0/1
			11219146	T	C	MOD	0/0	0/0	0/0	0/1	0/1
			11302080	T	TA	MOD	0/1	0/1	0/1	1/1	1/1
14	CDH20 (cadherin 20)	1	15018520	CTTTTTTTTTTTTT	C	MOD	0/0	0/0	0/0	0/1	0/1
			15026159	C	G	MOD	0/0	0/0	0/0	0/1	0/1
			15385740	C	T	MOD	0/0	0/0	0/0	0/1	0/1
			15397278	C	T	MOD	0/0	0/0	0/0	0/1	0/1
15	CDH23 (cadherin 23)	4	22217920	A	C	MOD	0/0	0/0	0/0	0/1	0/1
			22268845	G	A	MOD	0/0	0/0	0/0	0/1	0/1
			22296573	A	T	MOD	0/0	0/0	0/0	0/1	0/1
			22308343	G	C	MOD	0/0	0/0	0/0	0/1	0/1
			22521773	T	G	MOD	0/0	0/0	0/0	0/1	0/1
16	CHD1L (chromodomain helicase DNA-binding protein 1-like)	17	57755822	A	C	MOD	0/0	0/0	0/0	0/1	0/1
			57801618	C	T	MOD	0/0	0/0	0/0	0/1	0/1
17	CTNND2 (catenin delta 2)	34	2587961	C	G	MOD	0/0	0/0	0/0	0/1	0/1
			2605251	C	CAG	MOD	0/1	0/1	0/1	1/1	1/1
			2683554	C	CT	MOD	0/1	0/1	0/1	1/1	1/1
			2729501	C	G	MOD	0/0	0/0	0/0	0/1	0/1
			2790163	CAG	C	MOD	0/0	0/0	0/0	0/1	0/1
			2887443	G	T	MOD	0/0	0/0	0/0	0/1	0/1
			2926914	GA	G	MOD	0/0	0/0	0/0	0/1	0/1
			3386604	T	TTTC	MOD	0/1	0/1	0/1	1/1	1/1
			1984298	C	A	MOD	0/0	0/0	0/0	0/1	0/1
			2045599	A	AG	MOD	0/1	0/1	0/1	1/1	1/1
			2211983	T	TACACAC	MOD	0/1	0/1	0/1	1/1	1/1
			2325817	T	A	MOD	0/0	0/0	0/0	0/1	0/1
			2340793	C	G	MOD	0/0	0/0	0/0	0/1	0/1
			2420433	T	A	MOD	0/1	0/1	0/1	1/1	1/1
			2420448	G	A	MOD	0/1	0/1	0/1	1/1	1/1
18	DACT1 (Dishevelled binding antagonist of beta catenin 1)	8	33957483	G	A	HIGH	0/0	0/0	0/0	0/1	0/1
			33957480	C	T	LOW	0/0	0/0	0/0	0/1	0/1
19	EN1 (engrailed homeobox 1)	19	31181296	T	C	MOD	0/0	0/0	0/0	0/1	0/1
			31191156	TA	T	MOD	0/1	0/1	0/1	1/1	1/1
20	ERBB4 (Erb-b2 receptor tyrosine kinase 4)	37	19286254	T	G	MOD	0/0	0/0	0/0	0/1	0/1
			19629719	C	CT	MOD	0/1	0/1	0/1	1/1	1/1
			19767395	TTTC	T	MOD	0/1	0/1	0/1	1/1	1/1
			20133592	T	A	MOD	0/0	0/0	0/0	0/1	0/1
21	FBXW11 (F-box and WD repeat domain-containing 11)	4	40367311	CT	C	MOD	0/1	0/1	0/1	1/1	1/1
			40278734	T	TA	MOD	0/1	0/1	0/1	1/1	1/1
			40314614	C	T	MOD	0/0	0/0	0/0	0/1	0/1
22	FYN (tyrosine-protein kinase)	12	68231298	A	C	MOD	0/0	0/0	0/0	0/1	0/1
			68274356	T	TTA	MOD	0/1	0/1	0/1	1/1	1/1
23	FZD4 (frizzled class receptor 4)	21	12700289	G	T	MOD	0/0	0/0	0/0	0/1	0/1
24	FZD8 (frizzled class receptor 8)	2	1329228	T	G	MOD	0/0	0/0	0/0	0/1	0/1
			1333216	TA	T	MOD	0/0	0/0	0/0	0/1	0/1
25	INO80 (INO80 complex ATPase subunit)	30	8165408	A	T	MOD	0/0	0/0	0/0	0/1	0/1
			8165412	A	T	MOD	0/0	0/0	0/0	0/1	0/1
			8165416	A	T	MOD	0/0	0/0	0/0	0/1	0/1
			8206520	TA	T	MOD	0/0	0/0	0/0	0/1	0/1
26	ITPR2 (inositol 1,4,5-trisphosphate receptor type 2)	27	20831508	A	AG	MOD	0/1	0/1	0/1	1/1	1/1
			20872227	A	C	MOD	0/0	0/0	0/0	0/1	0/1
27	MAP3K7 (mitogen-activated protein kinase kinase kinase 7)	12	49749924	A	AT	MOD	0/0	0/0	0/0	0/1	0/1
			49769002	C	CT	MOD	0/1	0/1	0/1	1/1	1/1
28	PCDH10 (protocadherin 10)	19	8973713	T	C	MOD	0/0	0/0	0/0	0/1	0/1
			9008967	G	T	MOD	0/0	0/0	0/0	0/1	0/1
			9157583	C	T	MOD	0/0	0/0	0/0	0/1	0/1
29	PCDH11X (protocadherin-11 X-linked)	X	68666259	G	C	MOD	0/0	0/0	0/0	0/1	0/1
			68746381	G	A	MOD	0/0	0/0	0/0	0/1	0/1
30	PCDH15 (protocadherin-related 15)	26	34702015	T	C	MOD	0/0	0/0	0/0	0/1	0/1
			35097300	T	G	MOD	0/0	0/0	0/0	0/1	0/1
			33646637	G	A	MOD	0/0	0/0	0/0	0/1	0/1
			33672867	T	G	MOD	0/0	0/0	0/0	0/1	0/1
			33672897	G	A	MOD	0/0	0/0	0/0	0/1	0/1
31	PCDH18 (protocadherin 18)	19	5101338	G	A	MOD	0/0	0/0	0/0	0/1	0/1
			5101376	G	A	MOD	0/0	0/0	0/0	0/1	0/1
32	PCDH19 (protocadherin 19)	X	74412666	T	G	MOD	0/0	0/0	0/0	0/1	0/1
			74418677	G	A	MOD	0/0	0/0	0/0	0/1	0/1
			74250517	C	T	MOD	0/0	0/0	0/0	0/1	0/1
			74256962	G	A	MOD	0/1	0/1	0/1	1/1	1/1
			74260225	G	A	MOD	0/0	0/0	0/0	0/1	0/1
33	PCDH7 (protocadherin 7)	3	80140770	TAAA	T	MOD	0/1	0/1	0/1	1/1	1/1
			80356304	G	GA	MOD	0/1	0/1	0/1	1/1	1/1
			79602331	T	TGA	MOD	0/1	0/1	0/1	1/1	1/1
			79874335	T	A	MOD	0/0	0/0	0/0	0/1	0/1
			79951643	G	A	MOD	0/0	0/0	0/0	0/1	0/1
34	PCDH9 (protocadherin 9)	22	22259657	CA	C	MOD	0/0	0/0	0/0	0/1	0/1
			22267695	C	T	MOD	0/0	0/0	0/0	0/1	0/1
			20900771	T	A	MOD	0/0	0/0	0/0	0/1	0/1
			21327589	T	TA	MOD	0/1	0/1	0/1	1/1	1/1
			21447120	CAG	C	MOD	0/0	0/0	0/0	0/1	0/1
			21629025	T	A	MOD	0/0	0/0	0/0	0/1	0/1
			21682718	TA	T	MOD	0/0	0/0	0/0	0/1	0/1
			21741598	G	A	MOD	0/0	0/0	0/0	0/1	0/1
			21855444	T	G	MOD	0/0	0/0	0/0	0/1	0/1
			21977854	G	GA	MOD	0/1	0/1	0/1	1/1	1/1
35	PLCB1 (phospholipase C beta 4)	24	13687020	C	T	MOD	0/1	0/1	0/1	1/1	1/1
			13703533	CCCTCACACACA	C	MOD	0/1	0/1	0/1	1/1	1/1
			13506621	AT	A	MOD	0/0	0/0	0/0	0/1	0/1
			13545968	C	A	MOD	0/0	0/0	0/0	0/1	0/1
36	PLCB4 (phospholipase C beta 4)	24	13106067	G	A	MOD	0/0	0/0	0/0	0/1	0/1
			13222123	C	CT	MOD	0/1	0/1	0/1	1/1	1/1
37	PPP3CC (protein phosphatase 3 catalytic subunit gamma)	25	34764968	TA	T	MOD	0/0	0/0	0/0	0/1	0/1
			34812396	TA	T	MOD	0/0	0/0	0/0	0/1	0/1
38	PRKCA (protein kinase C alpha)	9	13673919	T	C	MOD	0/0	0/0	0/0	0/1	0/1
			13871044	TAA	T,TA	MOD	0/1	0/1	0/1	0/2	1/2
			13872388	G	GC	MOD	0/1	0/1	0/1	1/1	1/1
			13906274	G	A	MOD	0/0	0/0	0/0	0/1	0/1
			13963246	G	GT	MOD	0/0	0/0	0/0	0/1	0/1
			13987502	GA	G	MOD	0/0	0/0	0/0	0/1	0/1
39	PRKCE (protein kinase C epsilon type)	10	48018924	G	GT	MOD	0/1	0/1	0/1	1/1	1/1
			48021663	G	A	MOD	0/0	0/0	0/0	0/1	0/1
			48362949	A	T	MOD	0/1	0/1	0/1	1/1	1/1
			48362950	A	T	MOD	0/1	0/1	0/1	1/1	1/1
			48362951	A	T	MOD	0/1	0/1	0/1	1/1	1/1
			48452967	A	T	MOD	0/0	0/0	0/0	0/1	0/1
40	PRKCH (protein kinase C epsilon)	8	36326004	A	T	MOD	0/0	0/0	0/0	0/1	0/1
			36445470	C	A	MOD	0/0	0/0	0/0	0/1	0/1
41	PRKCQ (protein kinase C theta)	2	29254232	C	T	MOD	0/0	0/0	0/0	0/1	0/1
			29254236	C	T	MOD	0/0	0/0	0/0	0/1	0/1
			29246453	C	CA	MOD	0/1	0/1	0/1	1/1	1/1
42	TBL1X (transducin beta like 1 X-linke)	X	6446386	T	C	MOD	0/0	0/0	0/0	0/1	0/1
			6446390	T	C	MOD	0/0	0/0	0/0	0/1	0/1
43	TBXT (T-box transcription factor T)	1	53860297	T	G	MOD	0/0	0/0	0/0	0/1	0/1
			53860299	T	G	MOD	0/0	0/0	0/0	0/1	0/1
			53916908	G	C	MOD	0/0	0/0	0/0	0/1	0/1
			54140252	G	A	MOD	0/1	0/1	0/1	1/1	1/1
44	TCF7L2 (transcription factor 7 like 2)	28	23954797	A	C	MOD	0/0	0/0	0/0	0/1	0/1
			23973312	C	T	MOD	0/0	0/0	0/0	0/1	0/1
			24061992	C	T	MOD	0/0	0/0	0/0	0/1	0/1
45	TLE1 (TLE family member 1, transcriptional corepressor)	1	77969485	G	GT	MOD	0/1	0/1	0/1	1/1	1/1
			78272418	A	T	MOD	0/0	0/0	0/0	0/1	0/1
			77521907	A	AT	MOD	0/1	0/1	0/1	1/1	1/1
			77529973	CATAT	C	MOD	0/1	0/1	0/1	1/1	1/1
46	WNT5B (Wnt family member 5B)	27	43705981	A	T	MOD	0/0	0/0	0/0	0/1	0/1
			43718076	A	C	MOD	0/0	0/0	0/0	0/1	0/1
47	ENSCAFG00000008115	17	44082433	CTTTTTT	C,CTTTT	MOD	0/2	0/2	0/2	0/1	0/1
			44366744	T	C	MOD	0/0	0/0	0/0	0/1	0/1
			44392169	C	A	MOD	0/0	0/0	0/0	0/1	0/1
48	CTNNA3 (catenin alpha 3)	4	17723198	C	CTCTG	MOD	0/1	0/1	0/1	1/1	1/1
			18058494	T	C	MOD	0/0	0/0	0/0	0/1	0/1
			18069599	CT	C	MOD	0/0	0/0	0/0	0/1	0/1
			16891381	CA	C	MOD	0/1	0/1	0/1	1/1	1/1
			16989930	G	A	MOD	0/0	0/0	0/0	0/1	0/1
			16989932	A	T	MOD	0/0	0/0	0/0	0/1	0/1
			17291872	TA	T	MOD	0/0	0/0	0/0	0/1	0/1
49	ENSCAFG00000023180	X	99421688	G	A	MOD	0/0	0/0	0/0	0/1	0/1
			99421696	G	A	MOD	0/0	0/0	0/0	0/1	0/1
			99421697	G	T	MOD	0/0	0/0	0/0	0/1	0/1
			99421711	G	C	MOD	0/0	0/0	0/0	0/1	0/1
			99421728	T	C	MOD	0/0	0/0	0/0	0/1	0/1
			99421731	G	T	MOD	0/0	0/0	0/0	0/1	0/1
			99421735	C	T	MOD	0/0	0/0	0/0	0/1	0/1
			99421993	C	T	MOD	0/0	0/0	0/0	0/1	0/1
			99421995	C	T	MOD	0/0	0/0	0/0	0/1	0/1
			99422007	C	G	MOD	0/0	0/0	0/0	0/1	0/1
			99422009	A	C	MOD	0/0	0/0	0/0	0/1	0/1
50	LOC489647 (cadherin-1-like)	5	63556214	G	A	MOD	0/0	0/0	0/0	0/1	0/1
			63556216	G	A	MOD	0/0	0/0	0/0	0/1	0/1
			63556219	C	G	MOD	0/0	0/0	0/0	0/1	0/1
			63556227	T	C	MOD	0/0	0/0	0/0	0/1	0/1
			63556230	T	C	MOD	0/0	0/0	0/0	0/1	0/1
			63556233	T	C	MOD	0/0	0/0	0/0	0/1	0/1

CHR = chromosome, POS = position (genomic coordinate), REF = reference allele, ALT = alternative allele, ^+^ MOD = moderate.

## Data Availability

The data presented in this study are available from the corresponding author on reasonable request.

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
