# Peer review of "Brachygnathia Inferior in Cloned Dogs Is Possibly Correlated with Variants of Wnt Signaling Pathway Initiators"

_ijms, 2022, doi:10.3390/ijms23010475_

Round 1
Reviewer 1 Report
This interesting and well-written manuscript identified the potential genetic variants associated with the development of brachygnathia inferior in cloned dog animal subjects. While I have no doubts with the study design and that the authors have extensively discussed the results, there is some information lacking in the methods and results that needs to be addressed.
- I presume the authors would have tested all four cloned animal subjects in this study. Therefore, in both Figures 2 and 3, please incorporate also the clinical diagnosis and karyotyping results of both NT-3 and NT-4 animals. Otherwise, these results should at least be included as supplemental material.
- In Tables 2 and 3, the readings of NT-3 subject are missing. Please update. Additionally, why would the authors include the words “mean values” in the table title? Had multiple full blood counts and biochemical testings been done for each animal? If so, please indicate how many times the animals had been tested in the table footnote and the values should be presented as mean +/- standard deviation. In Table 2, please provide the full form of each abbreviation.
- In Table 4, I would suggest replacing the gene IDs with variant effects, and to focus on variants that had a moderate to high impact on the coding sequences including frameshift, nonsense and missense mutations.
- Table S2 is not necessary as the information in this table can also be found in both S1 and S3 tables.
- In section 4.7, please elaborate further the data processing steps. For examples, what software and parameters were used for quality checking and trimming of the raw sequence data and mapping of quality-trimmed reads to the reference genome? Also, Samtools is used for variant calling, while SnpEff is used to predict and annotate the effects of variants. Please revise.
- Please deposit the sequencing data into a public repository.
L76: at 1 > one
L95: 1 month > one month old
L115: please delete additional punctuation marks
L145: donor dog
L166: delete “, but no significant difference was found”
L167-168: Please consider the following correction: Coexisting variants present in all animal subjects including the donor dog were filtered.
L170: delete “expressed”
L171: [ⅱ - (ⅰ ⅱ)] > (Figure 4B)
L223: 30 were common genes between ⅱ and ⅲ of Figure 5D > 30 were shared between the Wnt and cadherin signaling pathways
L282-285: Can the authors please clarify if both n.5244256C>T and n.5248552A>G mutations are located in the CDS or intergenic region? A synonymous mutation can only occur in a CDS, but not in the intergenic region. If it turns out to be a synonymous mutation, it is very unlikely that there would be any detrimental effect leading to mandibular abnormalities. But, if the mutation happens to occur in an enhancer DNA sequence in the intergenic region, it is then difficult to evaluate the impact on mandibular abnormalities.
Author Response
Reviewer 1
- I presume the authors would have tested all four cloned animal subjects in this study. Therefore, in both Figures 2 and 3, please incorporate also the clinical diagnosis and karyotyping results of both NT-3 and NT-4 animals. Otherwise, these results should at least be included as supplemental material.
Answer: We added karyotyping results of NT-3 and NT-4 in figure 3. Result of clinical diagnosis was also added to supplementary figure 1.
- In Tables 2 and 3, the readings of NT-3 subject are missing. Please update. Additionally, why would the authors include the words “mean values” in the table title? Had multiple full blood counts and biochemical testings been done for each animal? If so, please indicate how many times the animals had been tested in the table footnote and the values should be presented as mean +/- standard deviation. In Table 2, please provide the full form of each abbreviation.
Answer: We added result of NT3 and full name of each abbreviation in table 3 and table 2 (line 226-228), respectively.
Title of table 2 and 3 were corrected from “mean values of “to “the values of”.
This study was conducted for a special purpose of the government, and NT 3 participated in a training program for that purpose. Therefore, NT3 could not perform automatic blood biochemical analysis, but only mechanical analysis (blood count, CBC). The results were attached as follows, and all values were within the normal range except for mild anemia to be due to chewing disorders.
Based on the review suggestion, we can insert the figure below as supplementary figure.
Figure for reviewer. Blood smear examination from blood cells of NT-3 (Diff-Quik stain). (A) Counting windows were determined. Red blood cells (RBCs) are monolayered, and cell clumping is not observed (i, × 200). Mild anisocytosis (8.8 cells/hpf), polychromasia (1.6 cells/hpf) were found on higher magnification. Mild morphological changes poikilocytosis (8.7 cells/hpf) including elliptocytes (3.4 cells/hpf), codocytes (3.8 cells/hpf) and acanthocytes (8.4 cells/hpf) were observed (ii, ×1,000). (B) Appropriate number of platelets were observed (13 platelets/hpf). Several platelets had pseudo pods and centralized granules. Black arrow indicates non-activated platelet. Magnification (×1,000). (C) Differential count of white blood cells (WBCs) revealed 15.76 × 10^9 mature neutrophils/L, 4.86 × 10^9 lymphocytes/L, 2.31 × 10^9 monocytes/L, and 0.02 × 10^9 eosinophils/L. Two metarubricytes out of 200 WBCs were observed (i, × 200). Neutrophils showed moderate number (24%) of mild cytoplasmic basophilia, Dohle bodies and foamy cytoplasm (×1,000). *hpf, high power field
- In Table 4, I would suggest replacing the gene IDs with variant effects, and to focus on variants that had a moderate to high impact on the coding sequences including frameshift, nonsense and missense mutations.
Answer: We inputted additional information in Table 4 by reviewer suggestion.
- Table S2 is not necessary as the information in this table can also be found in both S1 and S3 tables.
Answer: We deleted table S2 by reviewer comment
- In section 4.7, please elaborate further the data processing steps. For examples, what software and parameters were used for quality checking and trimming of the raw sequence data and mapping of quality-trimmed reads to the reference genome? Also, Samtools is used for variant calling, while SnpEff is used to predict and annotate the effects of variants. Please revise.
Answer: We have inserted a detailed analysis method as follows according to the comments of the review in 4.7 Whole-genome sequencing, sequence mapping, and variant calling (line 397-404).
“Skewer software (v0.2.2) were used for adapter trimmer, BWA (v0.7.15) were used for aligning the collected sequence data to the canine reference genome (CanFam 3.1). The Genome Analysis Toolkit (GATK, v2.3.9Lite) was used for improvement of alignment errors and genotype calling and refining with default parameters. SNP-calling procedure was performed to discover SNPs using SAM tools (v.1.3.1). The detected SNPs were then annotated to functional categories using SnpEff software (v4.3a).”
- Please deposit the sequencing data into a public repository.
Answer: We need a lot of process and time before putting a lot of WGS information on a public database (SRA). We were allowed a limited time of 10 days. Our research was carried out for a special purpose and disclosure of genetic information requires permission. If a researcher requests information for research, we may disclose some information.
- L76: at 1 > one
Answer: We changed from “at 1 day” to “one day” (line 77).
- L95: 1 month > one month old
Answer: We changed from “1 month” to “one month old” (line 97).
- L115: please delete additional punctuation marks
Answer: We deleted additional punctuation mark (line 118).
- L145: donor dog
Answer: We changed from “donor dogs” to “donor dog” (line 150).
- L166: delete “, but no significant difference was found”
Answer: We deleted “but no significant difference was found” (line 170-171).
- L167-168: Please consider the following correction: Coexisting variants present in all animal subjects including the donor dog were filtered.
Answer: We changed from “Coexisting variants present in all animal subjects including the donor dog were filtered” to“Coexisting variants present in all animal subjects including the donor dog were filtered (Figure 4B)”(line 171-172).
- L170: delete “expressed”
Answer: We deleted “expressed”in line 173.
- L171: [ⅱ - (ⅰ Ç ⅱ)] > (Figure 4B)
Answer: We followed reviewer suggestion (line 173).
- L223: 30 were common genes between ⅱ and ⅲ of Figure 5D > 30 were shared between the Wnt and cadherin signaling pathways
Answer: We changed from “30 were common genes between ⅱ and ⅲ of Figure 5D” to “30 were shared between the Wnt and cadherin signaling pathway” (line 226-227).
- L282-285: Can the authors please clarify if both n.5244256C>T and n.5248552A>G mutations are located in the CDS or intergenic region? A synonymous mutation can only occur in a CDS, but not in the intergenic region. If it turns out to be a synonymous mutation, it is very unlikely that there would be any detrimental effect leading to mandibular abnormalities. But, if the mutation happens to occur in an enhancer DNA sequence in the intergenic region, it is then difficult to evaluate the impact on mandibular abnormalities.
Answer: Really appreciate the review comment. we changed from “However, these two single nucleotide variants were a synonymous mutation and located in the intergenic region” to “However, these two single nucleotide variants were located in the intergenic region”.

Reviewer 2 Report
The authors report a dog cloning experiment, in which 2 out of 4 surviving cloned dogs showed brachygnathia inferior. The authors performed whole genome sequencing and claim that genomic variants in Wnt-related genes may have a role in causing the observed brachygnathia inferior phenotype. Unfortunately, the study is not well designed and the experimental data do not sufficiently support the far-reaching claims of the authors.
Specific comments:
(1)
The phenotypic characterization was well performed and represents an interesting finding. (brachygnathia is misspelled several times, including in figure 2.)
(2)
It can by no means taken as granted that the brachygnathia inferior phenotype is caused by genetic variation. In order to prove this, one would need to breed with the cloned dogs and only if the trait is transmitted to naturally produced offspring, a genetic mechanism is confirmed. I strongly suspect that the brachygnathia inferior was caused by reprogramming defects during the cloning process (=epigenetic mechanism).
(3)
The description of the WGS methodology is extremely superficial and partially incorrect (e.g. line 399: SnpEff is a software to annotate the functional effects of variants, it does not perform variant calling). Which annotation was used?
(4)
The WGS data must be deposited in a public database and accession numbers must be given.
(5)
I seriously doubt that the cloned dogs have so many genetic variants. I strongly suspect that the variant lists contain a very large proportion of false positive variant calls that are caused by technical artifacts. The authors should validate a subset of the detected variants with an independent method (e.g. PCR and Sanger sequencing).
(6)
The presentation of the genetic data is insufficient. Variants need to be annotated with their predicted functional effects on the protein level and the individual genotypes of all sequenced dogs must be given.
Author Response
Reviewer 2
- The phenotypic characterization was well performed and represents an interesting finding. (brachygnathia is misspelled several times, including in figure 2.)
Answer: We carefully checked and corrected all of misspelled ‘brachygnathia inferior’ in the manuscript and figures and marked with yellow highlighter or red text.
- It can by no means taken as granted that the brachygnathia inferior phenotype is caused by genetic variation. In order to prove this, one would need to breed with the cloned dogs and only if the trait is transmitted to naturally produced offspring, a genetic mechanism is confirmed. I strongly suspect that the brachygnathia inferior was caused by reprogramming defects during the cloning process (=epigenetic mechanism).
Answer: Thank you very much for your meaningful point regarding our study. We agree that meaningful results of the genetic mechanism can be obtained by confirmation of germline transmission of cloned dogs with brachygnathia inferior phenotype in further study.
Unfortunately, our dogs are working dogs not experimental dog and they are all neutered at the right time. For the same reason, the original dog was also neutered, so there was no choice but to clone it to obtain offspring. So, as mentioned in the last part of the conclusion, we will collect and analyze the results of dog with brachygnathia inferior in natural birth.
- The description of the WGS methodology is extremely superficial and partially incorrect (e.g. line 399: SnpEff is a software to annotate the functional effects of variants, it does not perform variant calling). Which annotation was used?
Answer: We have inserted a detailed analysis method as follows according to the comments of the review in 4.7 Whole-genome sequencing, sequence mapping, and variant calling (line 397-404).
“Skewer software (v0.2.2) were used for adapter trimmer, BWA (v0.7.15) were used for aligning the collected sequence data to the canine reference genome (CanFam 3.1). The Genome Analysis Toolkit (GATK, v2.3.9Lite) was used for improvement of alignment errors and genotype calling and refining with default parameters. SNP-calling procedure was performed to discover SNPs using SAM tools (v.1.3.1). The detected SNPs were then annotated to functional categories using SnpEff software (v4.3a).”
- The WGS data must be deposited in a public database and accession numbers must be given.
Answer: We need a lot of process and time before putting a lot of WGS information on a public database (SRA). We were allowed a limited time of 10 days. Our research was carried out for a special purpose and disclosure of genetic information requires permission. If a researcher requests information for research, we may disclose some information.
- I seriously doubt that the cloned dogs have so many genetic variants. I strongly suspect that the variant lists contain a very large proportion of false positive variant calls that are caused by technical artifacts. The authors should validate a subset of the detected variants with an independent method (e.g. PCR and Sanger sequencing).
Answer: Thank you for your insightful comment regarding our study. We agree that validation of variants was need to be performed. We are trying to verify the major genes related to the mandibular interior by PCR and Sanger sequencing. For example, FBXO5 was failed to get specific PCR product by a lot of poly T affecting RNA amplification. Wnt signaling genes, DACT1, ARRB2 and FZD4 also performed PCR as below but not obtained specific single PCR products for each gene. In addition, we are testing combination of various primers and Taq products but failed to generate a single PCR product for Sanger sequencing because of a high GC ratio to induce complex secondary structures, and poly T affecting RNA amplification.
If the reviewer suggests a good solution, we will try further analysis continuously. We will try to submit brief report on validation of variants by further study.
- The presentation of the genetic data is insufficient. Variants need to be annotated with their predicted functional effects on the protein level and the individual genotypes of all sequenced dogs must be given.
Answer: Thank you for critical insight and comments. In Table 4, the putative impact of each variant was added. In addition, individual genotypes of all sequenced dogs were also presented.

Reviewer 3 Report
Authors describe the findings of structural genetic variation associated with jaws skeleton abnormality in cloned dogs as the consequence of somatic cell nuclear transfer procedure (SCNT). Study includes the description of the applied procedure of SCNT cloning, occurred problems in obtaining of cloned dogs, genetic verification of their identity and the health condition including detailed clinical description of the diagnosed phenotype of brachygnathia inferior in two individuals. Next part of the manuscript includes the results of karyotyping and whole genome sequencing of samples derived from the donor of the nuclei and four SCNT cloned shepherd dogs among which two possessed abnormal shortening of the mandible. According to the presented results karyotyping showed no numerical abnormality of chromosomes nor chromosomal defects in investigated individuals pointing on the lack of large structural genomic abnormalities in inspected samples. However WGS allowed to reveal differences at the level of single nucleotide polymorphism mostly in genomic regions covering 3,164 genes. Most important results of gene ontology analysis included the gene significant enrichment in Wnt, cadherin, integrin signaling and gonadotropin-releasing hormone receptor pathways. Functional results relying on the protein-protein interactions allowed to extract 50 candidate genes for observed developmental defect being involved in both Wnt and cadherin signalling important for the proper bone formation.
It is interesting that not all cloned dogs suffered from described jaws defect. In the conclusion Authors underlie the possible role of epigenetic alterations during reprogramming of the donor somatic nuclei cells in the environment of foreign oocytes.
In my opinion presented study includes interesting findings contributing to the genetic background of developmental biology affected by somatic cloning. Going further It would be interesting to extend the research in the future on epigenetic alterations on the DNA methylation level on the genome scale.
Having some suggestions to the style, there is a lack of the separate Abbreviations and Conclusions sections in the article. Taking this fact into consideration, these lacking sections should have been added by the Authors at the end of the manuscript both to finally and comprehensively sum up the article (the summary sentences can be transferred from the end of Discussion section to the separate Conclusions section) and to explain/expand all the in-text abbreviations, which have been used by the Authors in each section of their manuscript.
Author Response
Reviewer 3
Having some suggestions to the style, there is a lack of the separate Abbreviations and Conclusions sections in the article. Taking this fact into consideration, these lacking sections should have been added by the Authors at the end of the manuscript both to finally and comprehensively sum up the article (the summary sentences can be transferred from the end of Discussion section to the separate Conclusions section) and to explain/expand all the in-text abbreviations, which have been used by the Authors in each section of their manuscript.
Answer: Thank you for your positive comments on our work. We carefully followed reviewer comments as follows: We inserted abbreviation and conclusion section in manuscript.

Round 2
Reviewer 2 Report
I fear that the minimal cosmetic edits are not sufficient to adress my serious concerns about this manuscript. I will briefly repeat my most important concerns:
(1) The authors did not consider and investigate the possibility that the brachygnathia inferior phenotype is caused by incorrect epigenetic reprogramming rather than by genetic differences between the clones. As epigenetic differences are much more likely than actual sequence differences in clones, this must be considered in the entire study design.
(2) The evidence for the claimed sequence differences between the clones is insufficient. Illumina short-read WGS is prone to false positive variant calls. At least some of the claimed sequence differences must be validated by an independent method.
(3) The raw data must be made publicly available. (Sequence accesisons for the WGS data)
Author Response
Respected reviewer 2
1) The authors did not consider and investigate the possibility that the brachygnathia inferior phenotype is caused by incorrect epigenetic reprogramming rather than by genetic differences between the clones. As epigenetic differences are much more likely than actual sequence differences in clones, this must be considered in the entire study design.
Answer: We made the corrections in response to reviewer concern (line 337-338).
(2) The evidence for the claimed sequence differences between the clones is insufficient. Illumina short-read WGS is prone to false positive variant calls. At least some of the claimed sequence differences must be validated by an independent method.
Answer: In the first revision note, we explained the detailed reason. Continuously, we are trying validation of variants by various ways. We are going to try to submit brief report on validation of variants through further study.
(3) The raw data must be made publicly available. (Sequence accesisons for the WGS data)
Answer:
: We have asked for government approval to upload vast amounts of WGS information in a public database (SRA). We need sufficient time for government approval and uploading of WGS. Our institute cannot access to SRA because of security barrier. So we must use external private serve to upload data. In addition, unknown errors often occur when data are uploaded on SGA. If we get government permission to disclose WGS information, we will try to register quickly.

Round 3
Reviewer 2 Report
The authors have not adressed my major concerns. It is unfortunate that this manuscript is now sent back and forth every other week without substantial changes. I am not willing to perform a thorough re-review without seeing the additional experiments requested in my first and second review.